# Optimization of Feed Components to Improve *Hermetia illucens* Growth and Development of Oil Extractor to Produce Biodiesel

**DOI:** 10.3390/ani11092573

**Published:** 2021-09-01

**Authors:** Kyu-Shik Lee, Eun-Young Yun, Tae-Won Goo

**Affiliations:** 1Department of Pharmacology, College of Medicine, Dongguk University, Gyeongju 38766, Korea; there1@dongguk.ac.kr; 2Department of Integrative Bio-Industrial Engineering, Sejong University, Seoul 05006, Korea; yuney@sejong.ac.kr; 3Department of Biochemistry, College of Medicine, Dongguk University, Gyeongju 38766, Korea

**Keywords:** biodiesel, black soldier fly (*Hermetia illucens*), organic waste, feed optimization, oil extractor

## Abstract

**Simple Summary:**

This investigation was performed to establish an optimal feed for *Hermetia illucens* (black soldier fly) larvae (HIL) and to develop an oil extractor for biodiesel production. An optimal feed for HIL for biodiesel production was established using organic wastes such as dried-food waste, chicken manure, and waste cooking oil. In addition, an automatic oil extractor was developed that cost-effectively produced industrial biodiesel, livestock feed, and fertilizer from HIL. Consequently, this investigation can contribute to the establishment of industrial systems for biodiesel production using HIL.

**Abstract:**

HIL are useful in agriculture because they can be used as feed for livestock or fertilizer and can bioconvert organic wastes, such as food waste and human and animal manure to usable fertilizer. In addition, HIL are being studied as a source of biodiesel because of their high-fat content. However, their use for biodiesel production has not been fully adopted. Here, the results showed that survival, weight gains, and total dried weight were significantly enhanced when HIL were fed dried-food waste (DFW)/chicken manure (CM). Furthermore, increased weight gain was observed in HIL fed DFW containing 5 mL waste cooking oil (WCO) per 100 g and 1.2% (*v*/*w*) fermented effective microorganism (F-EM). Based on these results, we prepared experimental feeds containing DFW, CM, WCO, and F-EM to establish an optimal feed for biodiesel production. We found that FT-1-2, a feed prepared with 60 g DFW, 40 g CM, 2 mL WCO, and 0.8% F-EM (*v*/*w*), significantly enhanced fat content, weight gain, and total dried weight of HIL. Our results indicate FT-1-2 is a suitable feed to breed HIL for biodiesel production. We then developed an automatic oil extractor for biodiesel production. The yield of the oil extractor was higher than that of solvent extraction. The study shows FT-1-2 is an optimal HIL feed for biodiesel production and that the developed oil extractor is useful for the extraction of crude oil from HIL and for the harvesting of defatted HIL frass for livestock feed and fertilizer. Taken together, we established an optimized low-cost feed for HIL breeding and developed an automatic oil extractor for the production of biodiesel from HIL.

## 1. Introduction

Biodiesel is a renewable clean energy and is considered an environment-friendly alternative fuel for heavy duty trucks and farm tractors [1]. Many natural lipid components in vegetable oil, waste cooking oil, and animal fat can be used as sources of biodiesel. Although fungi, bacteria, algae, and microalgae can be used to produce biodiesel, oil-bearing plants are considered as main sources [2]. However, biodiesel is still not widely used as a primary fuel for engines because it is more expensive to produce than fossil fuels [3,4]. The production cost of biodiesel is mainly determined by the feedstocks and catalysts used. Almost all biodiesel (more than 95%) is produced from edible oil extracted from vegetables [5,6], but vegetable oil is no longer viewed as an economical option. Vegetables and their oils are widely used for cooking and in the food industry and the price of vegetables is increasing. Furthermore, the use of crops to produce biodiesel causes serious environmental problems, such as destruction of soil resources and forests [2]. Therefore, to prevent these environmental problems and reduce production costs, it is important to discover new feedstocks for biodiesel production.

The huge amount of municipal solid waste (more than 2.1 billion tons per annum) dumped by humans is a major cause of global environmental pollution, and food waste accounts for about 25–45% of total solid waste [7]. Therefore, strategies based on the treatment of food waste are viewed as potential means of overcoming environmental pollution. Food waste treatment using insects is considered an environmental and economically viable means of recycling. Insects can bioconvert food wastes and human and animal manures to fertilizers. Some investigators have shown that food wastes bioconverted by insects can be used as environmentally friendly fertilizer and can reduce pathogenic microbes and pesticides [8,9,10,11]. Insects are also being investigated as alternative feedstocks for biodiesel production because insect larvae have high fat levels (16–57.9%) [12]. Moreover, insects can convert organic waste into biomass, and thus, biodiesel production by insects is considered an attractive means of preventing environmental pollution and reducing the cost of biodiesel production [13,14,15]. In fact, many researchers have reported biodiesel can be produced from insects fed organic waste [1,16,17,18,19,20].

Black soldier fly (*Hermetia illucens*) larvae (HIL) are known to biodegrade various organic wastes, such as livestock, human manure, food, industrial wastes, and produce biomass, and fat [1,21,22,23,24]. Some investigators have evaluated the fat contents of HIL fed different organic wastes and assessed the feasibility of extracting oil from organic waste-fed HIL [1,20]. In addition to the use of HIL as a feedstock for biodiesel production, they are considered to be a source of antimicrobial peptides [25,26,27]. Moreover, HIL and its frass can be used as feed for livestock and fertilizer [28]. Consequently, investigations suggest that HIL may be useful in a wide range of eco-friendly fields such as organic waste recycling, biodiesel production, and natural antimicrobial manufacture.

Li et al., (2011) showed that biomass, crude fat, and biodiesel yields of HIL are determined by the type of organic waste fed [1]. Therefore, we tried to develop a feed based on organic wastes that improves the growth, increases the fat and oil contents, and reduces the cost of breeding HIL. In addition, we developed an oil extractor to effectively produce crude oil from HIL that does not require chemical treatment.

## 2. Materials and Methods

### 2.1. H. illucens Larvae

HIL were gifted by the Department of Agricultural Biology at the National Instituted of Agricultural Sciences of the Rural Development Administration (Wanju, Korea). HIL were grown under controlled conditions (26 ± 1 °C and 60% relative humidity).

### 2.2. Preparation of Organic Waste

Dried-food waste (DFW), waste cooking oil (WCO), and chicken manure (CM) were used as feed sources for HIL breeding and a fermented effective microorganisms mix (F-EM): a fermented mixed culture containing mainly yeast, lactic acid bacteria, and photosynthetic bacteria) provided by local government was added additives to suppress the smell caused by decomposition. DFW was obtained from a food waste dump, WCO from a restaurant, and CM from a chicken farm. To test the effects of feeds on growth rate and body weight gain ratio, we prepared feeds, such as DFW/WCO, DFW/CM, and DFW/F-EM, and determined differences in HIL growth and fat contents according to DFW to organic waste or F-EM ratios. Compositions are presented in Table 1. To determine weight gains, the weight change of each group was first determined according to the following formula: weight gain (mg) = total weight (mg) of HIL at 11 days—total weight (mg) of HIL at 0 day. Then, weight gains were converted to percentages by dividing weight change of each experimental group by that of control group.

We then prepared DFW/WCO/CM/F-ME mixtures as experimental feeds and observed HIL growth rates, body weight gain ratios, and fat contents to determine an optimal feed composition (Table 2).

### 2.3. HIL Breeding

Six-day-old HIL (0.3 mg/larva) were inoculated into feed prepared using organic wastes, grown for 11 days, and then harvested to measure fat content, nutritional composition, total dried weight, and weight gain.

### 2.4. Extraction of Crude Oil From HIL

HIL were dried in a 700 W microwave for 8.5 min to extract crude oil, which was extracted using an expeller press or by solvent extraction. To extract crude fat using the expeller press, dried HIL were pressed at 120 °C and the extracted oil was collected as crude oil, which was then filtered to remove HIL biomass. Nutritional and fatty acid compositions of crude fat were determined. For solvent extraction, dried HIL were mixed with same volumes of hexane, stirred for 12 h at 50 °C and then filtered to remove HIL frass. The hexane was then removed using a rotatory evaporator.

### 2.5. Analysis of HIL Nutritional and Fat Compositions

HIL body compositions were measured after drying in a microwave for 6 min. Analyses of crude protein, crude fat, crude ash, and moisture in HIL were performed at the Foundation of Agricultural Technology Commercialization and Transfer (Iksan, Korea). Fatty acids in crude lipid were analyzed by gas chromatography/mass spectrometry at the Korea Quality Testing Institute (Suwon, Korea).

### 2.6. Performance Evaluation of Oil Extraction

The performance of the developed oil extractor was evaluated by comparing it with than of solvent extraction. To evaluate performance, we measured the extraction yields and acid values of extracted oils. Oil extraction yields are presented as percentages of total crude fat in HIL.

### 2.7. Statistical Analysis

Data were analyzed by one-way analysis of variance followed by Tukey’s post hoc test using SPSS Ver. 20.0 (SPSS Inc., Chicago, IL, USA). Statistical significance was accepted for *p* values < 0.05. Results are presented as means ± SDs.

## 3. Results

### 3.1. Effect of DFW/CM on HIL Growth

Several investigators have reported that HIL growth and nutritional composition are determined by feed composition. Li et al. demonstrated the highest crude fat content in HIL was obtained when HIL were fed chicken manure. Therefore, we assessed the effect of DFW/CM on HIL survival, body weight gain and total dried weight. The results obtained showed highest weight gains for DFW/CM-2 (Figure 1B) and greatest dried HIL weights for DFW/CM-2 and -3 (Figure 1C). The lowest total dried HIL weights, survival numbers, and weight gains were observed when HIL were fed DFW/CM-5 (0 g DFW/100 g CM; Figure 1A,B).

### 3.2. Effect of DFW/ WCO on HIL Growth

Some investigators have reported that HIL fat profiles and contents and growth depend on lipid profiles in feed. Therefore, we estimated the effect of DFW/WCO on HIL survival, body weight gain and total dried weight. The results showed that the survival numbers of all HIL fed DFW/WCO were similar to that of control (Figure 2A). In contrast, total dried HIL weight and weight gains were higher for HIL fed DFW/WCO-4 and -5 (Figure 2B,C). However, DFW/WCO-1, -2, and -3 did not affect weight gains or total dried weight as compared with control.

### 3.3. Effect of DFW/F-EM on HIL Growth

Although organic waste is a useful feed for HIL breeding, it can be easily decomposed and has an unpleasant smell. To prevent the decomposition and suppress the smell, we used F-EM solution, which is a well-known eco-friendly material, and evaluated its impact on growth and weight gains of HIL fed DFW/F-EM. We found DFW/F-EM did not affect HIL survival (Figure 3A), but that DFW/F-EM-3 significantly increased HIL weight gain and total dried weight (Figure 3B,C).

### 3.4. Effect of DFW/CM/WCO/F-EM on HIL Growth

Based on the above results, we assessed the effect of DFW/CM/WCO/F-EM on HIL growth to optimize organic waste feed for HIL breeding. As shown in Table 2, we prepared three different types of feeds. Initially, we assessed the effect of feed type-1 (FT-1) on HIL growth. As shown in Figure 4B,C, higher weight gains and total dried-HIL weights were observed in HIL fed FT-1-1, FT-1-2, FT-1-3, or FT-1-4, but not FT-1-5, than in HIL fed DFW-only (control). HIL survival was unaffected by FT-1 feeds, except FT-1-5, which reduced HIL survival by 10.4% (Figure 4A). Highest weight gain (179.5%) was observed in HIL fed FT-1-2. FT-2 feeds, except FT-2-6, increased dried HIL total weight and weight gains without reducing HIL survival, and weight gain was highest for HIL fed FT-2-3 (169.2%; Figure 5). FT-3 feeds, except for FT-3-5, increased total dried-HIL weight and weight gain (166.1% in FT-3-4) (Figure 6).

### 3.5. Analysis of Nutritional and Fatty Acid Compositions of HIL

Based on the above results, we chose five different HIL fed DFW (the control), DFW/CM-3, DFW/WCO-5, DFW/E-FM-3, or FT-1-2, and analyzed nutritional parameters. As shown in Table 3, highest crude fat content was observed in HIL fed FT-1-2, which suggested FT-1-2 was the more suitable feed for biodiesel production. Then, we analyzed fatty acid compositions in crude fat extracted from HIL fed DFW or FT-1-2. The results showed that all fatty acids in FT-1-2-fed HIL, except myristoleic acid and plamitoleic acid, were higher than in DFW-fed HIL (Table 4).

### 3.6. Development of the Crude Oil Extractor Used to Extract Crude Fat from HIL

Oil extraction using organic solvents is a well-established method of improving oil recovery [29]. However, the organic solvents used must be removed to produce biodiesel, and residual organic solvents in defatted HIL frass must also be removed to produce livestock feed or fertilizer, because that can damage human and animal health, and the additional processing involved incurs costs. Therefore, we developed a crude oil extractor to extract crude fat from HIL for industrial biodiesel production. A schematic of the extractor is shown in Figure 7. Extraction yields obtained using the extractor and nutritional contents in extracted oils and HIL frass were evaluated. We found that the extraction yields were higher than those achieved by organic solvent extraction (Table 5). Furthermore, we confirmed the crude oil yield at 120 °C was higher than at 150 °C (Appendix A). Nutritional composition analysis showed that crude fat contents in oil were around 90% and that frass had a high crude protein content (64.00 ± 4.62%; Table 6). We also assessed the effect of the dry system on the extraction of crude oil. The result showed that higher crude fat content in the crude oil was observed when extracted microwave-dried HIL (Appendix A).

## 4. Discussion

Many investigators have concluded that HIL are useful for recycling food wastes and organic wastes and that HIL and their byproducts can be used as eco-friendly fertilizers, animal feeds, and protein sources [30,31,32,33]. Furthermore, we showed in a previous study that HIL powder infected with *Lactobacillus casei* can be used as a natural antimicrobial in feed and foods [25]. Recently, investigators have also shown HIL is a potent feedstock for biodiesel production [21,22,23,29]. However, no process has been established for producing biodiesel from HIL. Here, we established an optimal feed and developed an oil extractor for industrial biodiesel production from HIL. Some investigations have reported a relationship between fat content in HIL and feed composition [1,34,35,36], and that organic waste is a useful feed for breeding HIL and for producing biodiesel. Li et al. reported that HIL fat content was significantly increased when HIL were fed CM [1] but did not report the effect of CM on HIL survival or weight gain. In this investigation, we found that HIL fed CM-only had the lowest weight gains, survivals, and total dried weights regardless of whether WCO and/or F-EM (Figure 1 and Figure 4, Figure 5 and Figure 6), and that HIL fed DFW/CM showed the highest weight gains and total dried weights without any reduction in survival (Figure 1). These results show DFW/CM-based feeds are more suitable for breeding HIL than CM-only for biodiesel production.

In addition, we examined whether WCO and F-EM feeds increase HIL fat content. Although DFW/WCO-5 enhanced HIL weight gains (Figure 2), fat contents were unaffected (Table 3). In contrast, DFW/F-EM-3 increased HIL fat contents and enhanced weight gains and total dried weights (Table 3, Figure 3B,C). However, DFW/WCO and DFW/F-EM feeds, except DFW/WCO-5 and DFW/F-EM-3, did not affect weight gains or total dried weights. These results demonstrate that DFW feeds containing 1.2% F-EM increase HIL body fat contents over those of DFW-only but that DFW containing WCO does not.

Although DFW/WCO and DFW/F-EM feeds, except DFW/WCO-5 and DFW/F-EM-3, did not significantly enhance HIL weight gains or total dried weights, we consider that WCO could be used as a source of fat in HIL and that F-EM can be used to prevent the decomposition of organic wastes and reduce malodors. Interestingly, highest weight gains (191.9 ± 20.3%) and total dried weight (17.0 ± 1.8 mg) were achieved when HIL were fed FT-1-2 (Figure 4B,C). Furthermore, crude fat content in HIL fed FT-1-2 was the highest observed (Table 3). Although HIL total dried weights and crude fat contents were not significantly affected by DFW/WCO and DFW/F-EM, except DFW/WCO-5 and DFW/F-EM-3, our results indicate that WCO and F-EM enhance weight gain and dried weights in HIL fed FT-1-2. Furthermore, as shown by Table 4, almost all fatty acid contents were increased by feeding FT-1-2. Consequently, the results indicate that FT-1-2 is the optimal feed to breed HIL for biodiesel production.

Solvent extraction is generally considered the most effective means of biodiesel production from HIL, but as mentioned above these solvents must be removed from HIL frass to produce livestock feed and fertilizer, and this involves further costs. In contrast, mechanical extraction does not require further processing, although reported extraction yields are lower than those of solvent extraction [37]. In the present investigation, we established a high yield extraction method based on a high-performance extraction unit (Figure 7). As shown Table 5, when we extracted oil from microwave dried HIL, the yield obtained (37.23%) was greater than that of solvent extraction (26.97%). Furthermore, the mean fat content in extracted oil was 86.66 ± 1.54% and had a low moisture content (0.57 ± 0.15%) and the frass produced contained a low level of crude fat (7.17 ± 0.73) and had a high crude protein content (64.00 ± 4.62; Table 6). Also, fatty acid composition in crude fat extracted by expeller press was similar to that of chemical extraction reported by Surendra et al. (Table 4) [38]. These results indicate that the devised oil extractor provides an optimal means of extracting crude oil from HIL and for producing HIL frass for livestock feed and fertilizer.

## 5. Conclusions

In the present study, we established an optimal feed composition for breeding HIL for biodiesel production and developed a high yield oil extraction system. The study shows that FT-1-2 is an optimal feed for HIL breeding and biodiesel production and that the oil extractor provides a means of industrializing HIL for biodiesel, livestock feed, and fertilizer production, without the additional costs associated with solvent extraction.

## Figures and Tables

**Figure 1 animals-11-02573-f001:**
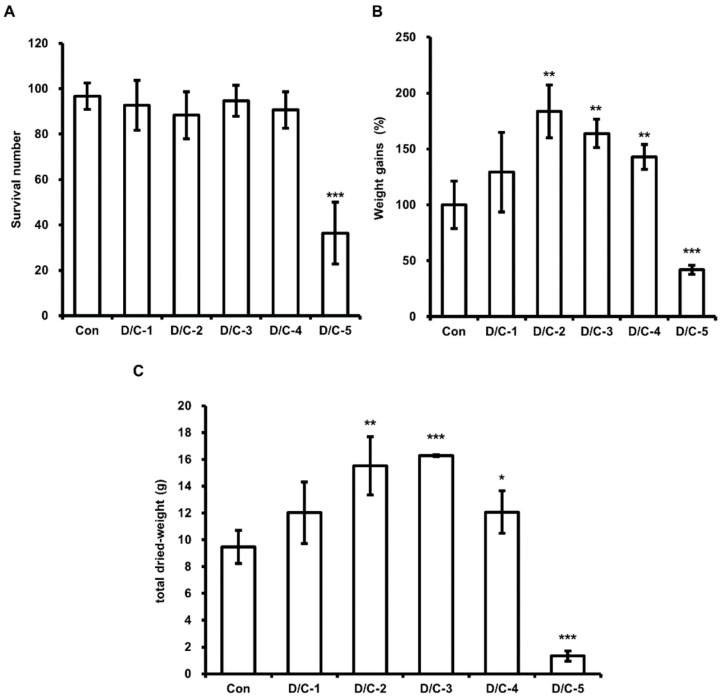
Evaluation of the effect of DFW/CM on HIL survival, weight gain, and total dried weight. (**A**) Surviving HIL were harvested and counted at 11 days after inoculating 6-day-old HIL. (**B**) Total weights were measured at baseline and 11 days after inoculation. The total weights of HIL at 11 days were divided by those at baseline. Relative differences are presented as weight gains (%). (**C**) Harvested HIL were dried using a microwave and total weights were measured. All experiments were independently performed in triplicate. Results are presented as means ± SDs; * *p* < 0.05, ** *p* < 0.01, *** *p* < 0.001.

**Figure 2 animals-11-02573-f002:**
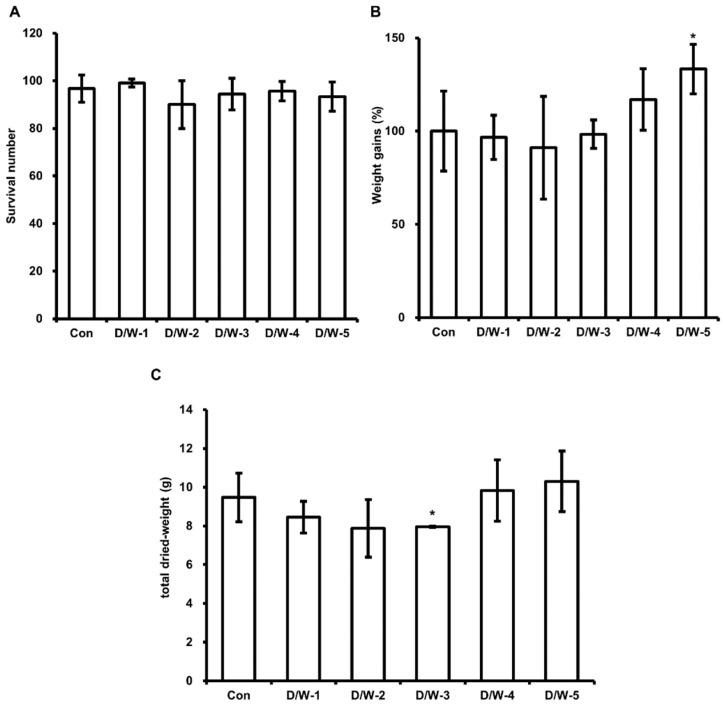
Evaluation of the effect of DFW/WCO on HIL survival, weight gain, and total dried weight. (**A**) Surviving HIL were harvested and counted at 11 days after inoculating 6-day-old HIL. (**B**) Total weights were measured at baseline and 11 days after inoculation. Total HIL weights at 11 days were divided by those at baseline, and relative differences are presented as weight gains (%). (**C**) Harvested HIL were dried using a microwave and total weights were measured. All experiments were independently performed in triplicate. Results are presented as means ± SDs; * *p* < 0.05.

**Figure 3 animals-11-02573-f003:**
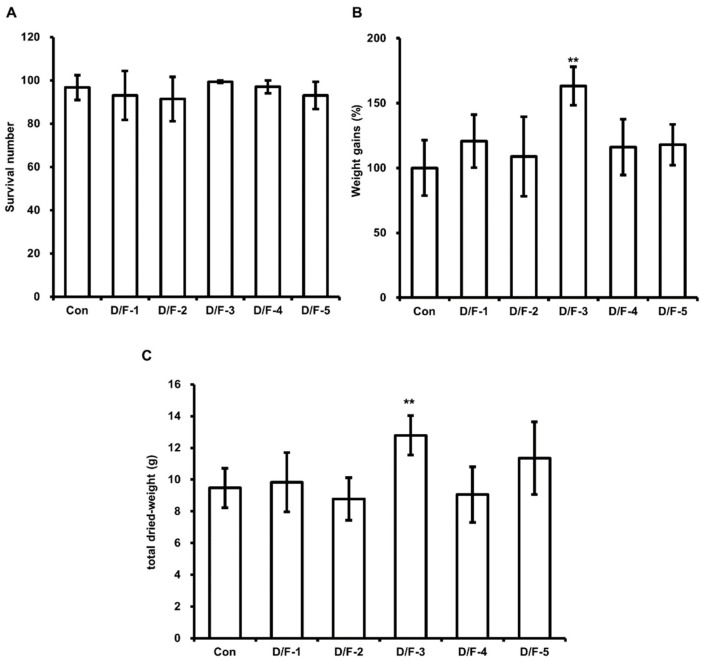
Evaluation of the effect of DFW/F-EM on HIL survival, weight gain, and total dried weight. (**A**) Surviving HIL were harvested and counted at 11 days after inoculating 6-day-old HIL. (**B**) Total weights were measured at baseline and 11 days after inoculation. Total weights of HIL at 11 days were divided by those at baseline, and relative differences are presented as weight gains (%). (**C**) Harvested HIL were dried using a microwave and total weights were measured. All experiments were independently performed in triplicate. Results are presented as means ± SDs; ** *p* < 0.01.

**Figure 4 animals-11-02573-f004:**
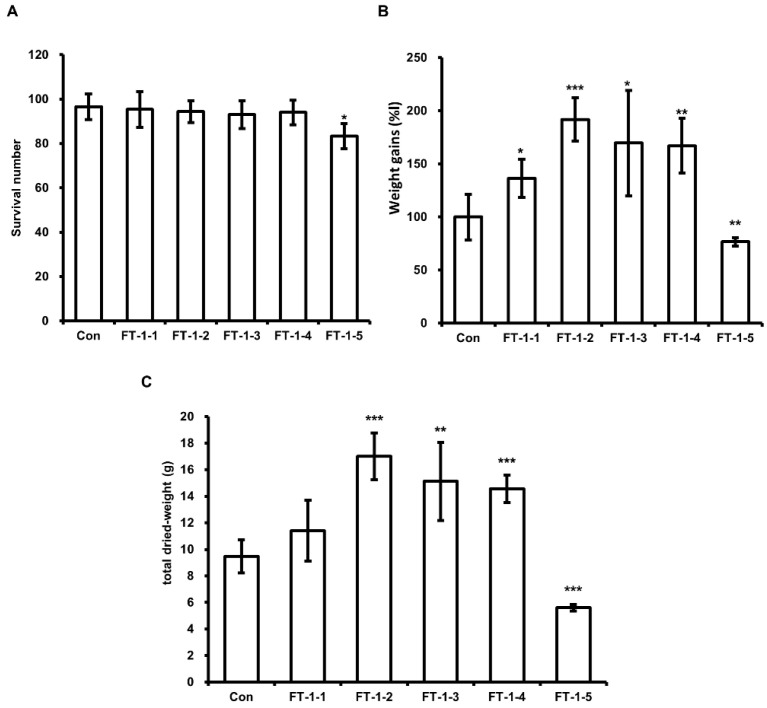
Evaluation of the effect of DFW/FT-1 feeds on HIL survival, weight gain, and total dried weight. (**A**) Surviving HIL were harvested and counted at 11 days after inoculating 6-day-old HIL. (**B**) Total weights were measured at baseline and 11 days after inoculation. Total weights of HIL at 11 days were divided by baseline weights. Relative differences are presented as weight gains (%). (**C**) Harvested HIL were dried using a microwave and then total weights were measured. All experiments were independently performed in triplicate. Results are presented as means ± SDs; * *p* < 0.05, ** *p* < 0.01, *** *p* < 0.001.

**Figure 5 animals-11-02573-f005:**
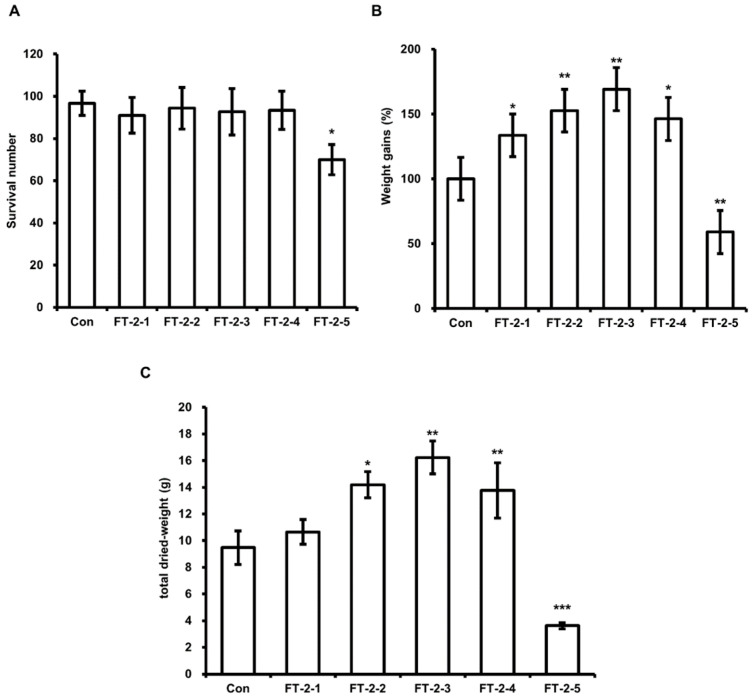
Evaluation of the effect of DFW/FT-2 feeds on HIL survival, weight gain, and total dried weight. (**A**) Surviving HIL were harvested and counted at 11 days after inoculating 6-day-old HIL. (**B**) Total weights were recorded at baseline and 11 days after inoculation. The total weights of HIL at 11 days were divided by baseline weights. The relative differences are presented as weight gains (%). (**C**) Harvested HIL were dried using a microwave and then total weights were measured. All experiments were independently performed in triplicate. Results are presented as means ± SDs; * *p* < 0.05, ** *p* < 0.01, *** *p* < 0.001.

**Figure 6 animals-11-02573-f006:**
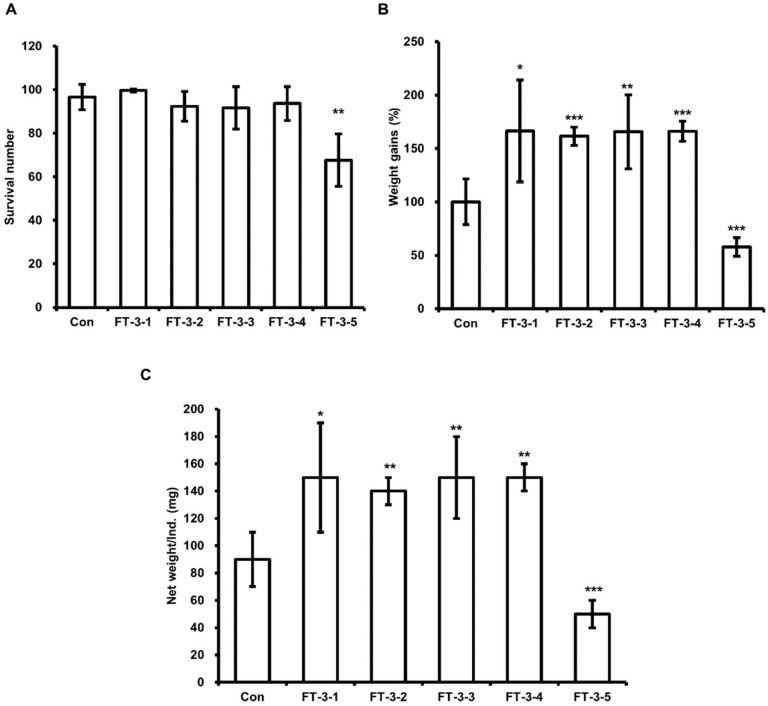
Evaluation of the effect of DFW/FT-3s on HIL survival, weight gain, and total dried weight. (**A**) Surviving HIL were harvested and counted at 11 days after inoculating 6-day-old HIL. (**B**) Total weights were measured at baseline and 11 days after inoculation. Total weights of HIL at 11 day were divided by baseline weights. Relative differences were presented as weight gains (%). (**C**) Harvested HIL were dried using a microwave and total weights were measured. All experiments were independently performed in triplicate. Results are presented as means ± SDs; * *p* < 0.05, ** *p* < 0.01, *** *p* < 0.001.

**Figure 7 animals-11-02573-f007:**
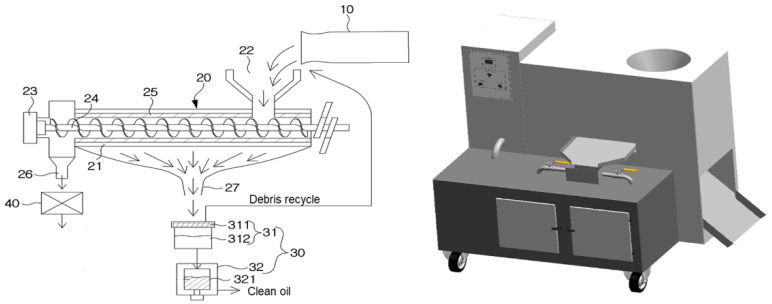
The schematic diagram and 3D of the oil extractor. (10) preheater, (20) extraction region, (21) housing, (22) larva input, (23) driven rotor, (24) rotatory screw, (25) compression bar, (26) frass outlet, (27) oil outlet, (30) oil purifying region, (31) primary filter assembly, (32) secondary filter assembly, (311) filter mesh, (312) oil collector, (321) high-performance filter, (40) frass grinder.

**Table 1 animals-11-02573-t001:** Components of DFW/WCO, DFW/CM, and DFW/F-EM feeds.

No.	DFW/CM		DFW/WCO		DFW/F-EM	
	DFW (g)	CM (g)	DFW (g)	WCO (mL)	DFW (g)	F-EM (%)
Control	100	0	100	0	100	0
1	80	20	100	1	100	0.4
2	60	40	100	2	100	0.8
3	40	60	100	3	100	1.2
4	20	80	100	4	100	1.6
5	0	100	100	5	100	2.0

**Table 2 animals-11-02573-t002:** Components of the experimental feeds prepared to optimize feed composition.

Feed Type	Sample Number	Components			
		DFW (g)	CM (g)	WCO (mL)	F-EM (%)
FT *-1	Control	100	0	0	0
	1	80	20	1	0.4
	2	60	40	2	0.8
	3	40	60	3	1.2
	4	20	80	4	1.6
	5	0	100	5	2.0
FT-2	Control	100	0	0	0
	1	80	20	5	0.4
	2	60	40	4	0.8
	3	40	60	3	1.2
	4	20	80	2	1.6
	5	0	100	1	2.0
FT-3	Control	100	0	0	0
	1	80	20	5	2.0
	2	60	40	4	1.6
	3	40	60	3	1.2
	4	20	80	2	0.8
	5	0	100	1	0.4

* FT: Feed type.

**Table 3 animals-11-02573-t003:** Comparison of nutritional compositions of HIL fed DFW, DFW/CM-3, DFW/WCO-5, DFW/F-EM-3, or FT-1-2.

Composition	Feed				
	DFW	DFW/CM-3	DFW/WCO-5	DFW/F-EM-3	FT-1-2
Moisture	9.34%	10.46%	8.84%	9.73%	11.28%
Crude protein	36.72%	35.91%	35.91%	36.12%	35.56%
Crude fat	29.50%	32.03%	30.85%	33.00%	33.87%
Crude ash	11.31%	7.95%	10.59%	8.11%	8.03%

**Table 4 animals-11-02573-t004:** Comparison of fatty acid composition in HIL fed DFW or FT-1-2.

Fatty Acid	Common Name	Unit	Feed	
			DFW	FT-1-2
C4:0	Butyric acid	g/100 g	n.d	n.d
C6:0	Caproic acid	g/100 g	0.0001	0.0003
C8:0	Caprylic acid	g/100 g	0.001	0.002
C10:0	Capric acid	g/100 g	0.15	0.21
C11:0	Undecanoic acid	g/100 g	0.002	0.003
C12:0	Lauric acid	g/100 g	2.6	3.4
C13:0	Tridecanoic acid	g/100 g	0.002	0.003
C14:0	Myristic acid	g/100 g	0.46	0.55
C14:1	Myristoleic acid	g/100 g	0.011	0.008
C15:0	Pentadecanoic acid	g/100 g	0.012	0.024
C15:1	cis-10-Pentadecenoic acid	g/100 g	n.d	n.d
C16:0	Palmitic acid	g/100 g	1.46	2.36
C16:1	Palmitoleic acid	g/100 g	0.18	0.15
C17:0	Heptadecanoic acid	g/100 g	0.015	0.027
C17:1	Cis-10-Heptadecanoic acid	g/100 g	0.009	0.012
C18:0	Stearic acid	g/100 g	0.29	0.32
C18:1	Oleic acid	g/100 g	1.04	1.57
C18:2	Linoleic acid	g/100 g	1.02	1.64
C18:3n-3	α-Linolenic acid	g/100 g	0.001	0.007
C18:3n-6	γ-Linolenic acid	g/100 g	n.d	0.002
C20:0	Arachidic acid	g/100 g	n.d	0.012
C20:1	cis-11-Eicosenoic acid	g/100 g	n.d	n.d
C20:2	cis-11,14-eicosadienoic acid	g/100 g	0.001	0.002
C20:3n-3	cis-11,14,17-Eicosadienoic acid	g/100 g	0.02	0.037
C20:3n-6	cis-8,11,14-Eicosadienoic acid			
C20:4n-6	Arachidonic acid	g/100 g	0.018	0.025
C20:5n-3	cis-5,8,11,14,17-Eicosapentaenoic acid	g/100 g	0.04	0.05
C21:0	Heneicosanoic acid	g/100 g	n.d	n.d

n.d: not detected.

**Table 5 animals-11-02573-t005:** Oil extraction yields of the oil extractor and solvent extraction.

	Extraction Method	
	Oil Extractor	Solvent Extraction
Yield (%)	37.23%	26.97%

**Table 6 animals-11-02573-t006:** Comparison of the nutritional compositions of crude oil and HIL frass.

Composition	Content (%)	
	Crude Oil	HIL Frass
Moisture	0.57 ± 0.15	1.19 ± 0.24
Carbohydrate	2.84 ± 0.87	20.90 ± 4.66
Crude protein	8.63 ± 0.98	64.00 ± 4.62
Crude fat	86.66 ± 1.54	7.17 ± 0.73
Crude ash	1.30 ± 1.62	10.67 ± 4.60

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
