# Peer review of "Optimization of Feed Components to Improve Hermetia illucens Growth and Development of Oil Extractor to Produce Biodiesel"

_animals, 2021, doi:10.3390/ani11092573_

Round 1
Reviewer 1 Report
Dear Authors,
Please find the attached review report.
Regards

Author Response
Thanks for sharpen your comments. We replied for your comments as below.
The paper generally reads well but I have just a few comments:
Line 5 – 11: Remove the hyphens or full stops or spaces from the initials of the authors. Do the same on the Authors contribution part. There is also no need to add TWG’s email twice.
: Thanks for your comments. We corrected completely.
Line 12 – 15: The simple summary can be improved with more details. Authors have a maximum of 200 words, so why not? For example, line 16 – 19 (overlapping to line 20) can be easily placed in the simple summary to avoid having a long abstract.
: Thanks for your comments. We corrected the simple summary and abstract.
Line 20: Replace ‘we tried to establish…’ with ‘The study aimed at establishing...’. Do something similar in line 77. Authors should avoid writing in second person terms.
: Thanks for you comments. We corrected the sentence to avoid writing in second person terms.
Line 53: Is that a national or global stat? If national, then name the country
: It is global stat. We added “global” in the sentence to prevent confusion.
Line 82: Write Hermetia in full when starting a sentence. Do the same with HIL
: Thanks for your comment. We corrected the sentences.
Line 83 and 84: Don’t start sentences with acronyms or abbreviations. Do the same throughout the text. Also correct ‘(Wanju), Korea)’
: Thanks for your comment. We corrected the sentence. ‘(Wanju), Korea)’ was corrected as ‘(Wanju, Korea)’.
Line 88: replace microorganism with microbial
: Thanks for your comment. Microorganism was corrected as microbial.
Line 89: replace lactic acid bacteria with lactate or lactic acid producing bacteria
: Thanks for your comment. Lactic acid bacterial was corrected as lactic acid producing bacteria.
Line 93: Mention how body weight gain ratio was calculated
: Thanks for your comment. We added how body weight gain ratio was calculated in the section.
Table 2: Don’t bold FT-1
: Thanks for your comment. We corrected.
The authors should not discuss or give literature/references in the results section. This causes unnecessary repetition of words in the introduction and discussion part.
: Thanks for your comment. We removed discussion and references in Result sections.
Reviewer 2 Report
The paper investigated the effect of the various feedstocks on the fat contents of Hermetia illucens (black soldier fly) larvae for biodiesel production. At last, the automatic oil extractor was developed to produce crude oil from the insect. There are several issues needed to be improved or modified and revised before the possible publication.
- Written English needs to be carefully revised along the manuscript.Some but not all are as the following:
Line 68, “such as livestock and human manure and food and industrial wastes, and produce biomass and fat” was changed with “such as livestock, human manure, food, industrial wastes, produce biomass, and fat.”
Line 138, “These results show DFW/CM-2 or -3 provided best HIL growth.” was changed with “These results showed DFW/CM-2 or -3 provided best HIL growth.”
Line 183, “ As shown in Figs. 4B and C” was changed with “As shown in Fig. 4 (B and C)”.
Line 262, “ When we investigated this topic we found HIL fed CM-only with/without WCO and F-EM had the lowest weight gains, survivals, and total dried-weights” what means?
- This article is written very carelessly as result serious errors in words. (for example: Line297, high crude protein content is 64±0.4.62 ? )
- Line 67, “Hermetia illucens”appears in italics in this paper.
- The paper presents a method to extract oil from the insect. If this method was an innovation in this paper. I think this section should be studied in detail.
For example,
- The effect of the different factors (microwave temperature, press time, press temperature, pressure) on the yield and fatty acid composition was studied.
- The microwave heating method was chose in this paper, why? The traditional heating method (drying oven) was OK?
- The temperature has a great influence on the fatty acid composition of oil. The temperature was 120 ℃in this paper. Did this temperature destroy the fatty acid composition of oil? The fatty acid composition of oil extracted by microwave and solvent should be compared.
Author Response
Thanks for sharpen your comments. We replied for your comments as below.
The paper investigated the effect of the various feedstocks on the fat contents of Hermetia illucens (black soldier fly) larvae for biodiesel production. At last, the automatic oil extractor was developed to produce crude oil from the insect. There are several issues needed to be improved or modified and revised before the possible publication.
Written English needs to be carefully revised along the manuscript. Some but not all are as the following:
Line 68, “such as livestock and human manure and food and industrial wastes, and produce biomass and fat” was changed with “such as livestock, human manure, food, industrial wastes, produce biomass, and fat.”
: Thanks for your comment. we corrected the sentence as you mentioned.
Line 138, “These results show DFW/CM-2 or -3 provided best HIL growth.” was changed with “These results showed DFW/CM-2 or -3 provided best HIL growth.”
: Thanks for your comment. we corrected the sentence as you mentioned.
Line 183, “As shown in Figs. 4B and C” was changed with “As shown in Fig. 4 (B and C)”.
: Thanks for your comment. we corrected the sentence as you mentioned.
Line 262, “When we investigated this topic we found HIL fed CM-only with/without WCO and F-EM had the lowest weight gains, survivals, and total dried-weights” what means?
: Thanks for your comment. We corrected the sentence to “In this investigation, we found that HIL fed CM-only had the lowest weight gains, survivals, and total dried-weights regardless of whether WCO and/or F-EM”
This article is written very carelessly as result serious errors in words. (for example: Line297, high crude protein content is 64±0.4.62?)
: Thanks for your comment. We corrected the errors.
Line 67, “Hermetia illucens” appears in italics in this paper.
: Thanks for your comment. We corrected “Hermetia illucens” to “Hermetia illucens”.
The paper presents a method to extract oil from the insect. If this method was an innovation in this paper. I think this section should be studied in detail.
For example,
- The effect of the different factors (microwave temperature, press time, press temperature, pressure) on the yield and fatty acid composition was studied.
: Thanks for your comments. Microwave was used for drying HIL. Furthermore, the incubation time is 6 min. At 2018, Charuwat et al. showed the effect of temperature on degradation of long chain fatty acids (Water Environ. Res., 2018, 90(3): 278-287). The result showed that saturated fatty acid was hydrolyzed when incubate at 140-160℃ for 8 h. It means fatty acid composition is not affected by your commented factors. Therefore, we didn’t assess the effect of the commented factors on fatty acid composition.
Expeller press is commonly used in oil extraction. We thought the pressure was similar to that of commercial extractors. Therefore, we assessed the effect of press temperature on the yield (Table S1 A). The result showed that the yield at 120℃ was higher than the yield at 150℃. Although we did not show the yield at low temperature, we found that extraction process was prevented by insoluble fatty acid.
The microwave heating method was chosen in this paper, why? The traditional heating method (drying oven) was OK?
: Thanks for your comments. Microwave was not used as heating method. It was used to dry HIL. Drying oven also can use to dry HIL. However, the drying of drying oven time (at least 6 h) was longer than that of microwave (6 min at 18 kw). We also found the higher crude fat content in crude oil extracted from microwave-dried HIL than that in drying oven-dried HIL (Table S2). Therefore, we chose microwave to produce industrial scale of crude fat from HIL.
- The temperature has a great influence on the fatty acid composition of oil. The temperature was 120 ℃ in this paper. Did this temperature destroy the fatty acid composition of oil? The fatty acid composition of oil extracted by microwave and solvent should be compared.
: Thanks for your comment. We used microwave shortly to dry HIL. Therefore, we think microwave does not affect to fatty acid composition in this investigation.
The fatty acid composition in oil extracted by solvent was already reported by several investigations. Therefore, we compared the fatty acid composition of oil extracted by expeller press with reported composition in oil extracted by solvent (Surendra et al., Renewable Energy 98 (2016) 197-202). We found the composition of fatty acid in crude oil extracted by expeller press was similar to that of solvent extraction. We discussed the results in Discussion section.